# The incremental value of the contribution of a biostatistician to the reporting quality in health research—A retrospective, single center, observational cohort study

Ulrike Held[1]*, Klaus Steigmiller[1], Michael Hediger[2], Victoria L. Cammann[3], Alexandru Garaiman[4], Sascha Halvachizadeh[5], Sylvain Losdat[6], Erin Ashley West[7], Martina Gosteli[8], Kelly A. Reeve[1], Stefanie von Felten[1], Eva Furrer[1]

1 Department of Biostatistics at Epidemiology, Biostatistics and Prevention Institute, University of Zurich, Zurich, Switzerland, 2 Institute of Mathematics, University of Zurich, Zurich, Switzerland, 3 Department of Cardiology, University Heart Center, University Hospital Zurich, University of Zurich, Zurich, Switzerland, 4 Department of Rheumatology, University Hospital Zurich, University of Zurich, Zurich, Switzerland, 5 Department of Trauma, University Hospital Zurich, University of Zurich, Zurich, Switzerland, 6 CTU Bern, University of Bern, Bern, Switzerland, 7 Department of Epidemiology at Epidemiology, Biostatistics and Prevention Institute, University of Zurich, Zurich, Switzerland, 8 Main Library, University of Zurich, Zurich, Switzerland

☯ These authors contributed equally to this work.
* ulrike.held@uzh.ch

## Abstract

### Background

The reporting quality in medical research has recently been critically discussed. While reporting guidelines intend to maximize the value from funded research, and initiatives such as the EQUATOR network have been introduced to advance high quality reporting, the uptake of the guidelines by researchers could be improved. The aim of this study was to assess the contribution of a biostatistician to the reporting and methodological quality of health research, and to identify methodological knowledge gaps.

### Methods

In a retrospective, single center, observational cohort study, two groups of publications were compared. The group of exposed publications had an academic biostatistician on the author list, whereas the group of non-exposed publications did not include a biostatistician of the evaluated group. Rating of reporting quality was done in blinded fashion and in duplicate. The primary outcome was a sum score based on six dimensions, ranging between 0 (worst) and 11 (best). The study protocol was reviewed and approved as a registered report.

### Results

There were 131 publications in the exposed group published between 2017 and 2018. Of these, 95 were either RCTs, observational, or prediction / prognostic studies. Corresponding matches in the group of non-exposed publications were identified in a reproducible manner.

**Data Availability Statement:** The data underlying
the results presented in the study are available
on the Open Science Framework (https://osf.io/
5egrn/).

**Funding:** The study was supported by the Center
for Reproducible Science at the University of Zurich
(https://www.crs.uzh.ch/en.html). The funder had
no role in study design, data collection and
analysis, decision to publish, or preparation of the
manuscript.

**Competing interests:** I have read the journal's
policy and SL has the following competing
interests: SL is employed by CTU Bern, University
of Bern, which has a staff policy of not accepting
honoraria or consultancy fees. However, CTU Bern
is involved in design, conduct, or analysis of
clinical studies funded by not-for profit and for-
profit organizations. In particular, pharmaceutical
and medical device companies provide direct
funding to some of these studies. For an up-to-date
list of CTU Bern's conflicts of interest: http://www.
ctu.unibe.ch/research/declaration of interest/index
eng.html. UH, KS, MH, VLC, AG, SH, EAW, MG,
KAR, SVF, and EF declare to have no conflict of
interest. This does not alter our adherence to PLOS
ONE policies on sharing data and materials.

Comparison of reporting quality overall revealed a 1.60 (95%CI from 0.92 to 2.28, p
<0.0001) units higher reporting quality for exposed publications. A subgroup analysis within
study types showed higher reporting quality across all three study types.

## Conclusion

Our study is the first to report an association of a higher reporting quality and methodological
strength in health research publications with a biostatistician on the author list. The higher
reporting quality persisted through subgroups of study types and dimensions. Methodologi-
cal knowledge gaps were identified for prediction / prognostic studies, and for reporting on
statistical methods in general and missing values, specifically.

## Introduction

Despite measures to increase the reporting quality in the field of health research, for example,
by introducing reporting guidelines and inclusion of such guidelines in recommendations for
authors by many publishers, quality standards are still oftentimes not met. Recent evaluations
of the literature showed that for observational studies, the corresponding STROBE guideline
was not used by nearly 18% of the authors because the authors had not heard of the guideline
before. An additional 19% of authors had heard of it but still did not use it [1]. Journals obvi-
ously play an important role, and a systematic evaluation showed that journal endorsement
rates to the STROBE guidelines are only around 50% [2]. When it comes to the reporting of
randomized trials, Dechartres et al. [3] have systematically evaluated reporting of more than
20'000 trials included in Cochrane reviews. They conclude that poor reporting has decreased
over time, but that especially lower impact factor journals show room for improvement.
Reporting quality of clinical prediction models has recently been evaluated systematically in
the context of research on Severe acute respiratory syndrome coronavirus 2 (Sars-Cov-2) [4].
The authors concluded that almost all published models for predicting mortality were poorly
reported, and that the corresponding Transparent reporting of a multivariable prediction
model for individual prognosis or diagnosis (TRIPOD [5]) guideline was largely omitted.

In Switzerland, the government paid 22.9 billion Swiss francs for research and develop-
ment, representing more than 3% of the gross domestic product in 2019. Publications in the
field of "clinical medicine" represent 25% of all publications [6], and given the large amounts
of resources, value from research and publications should be maximized.

The objectives of the current study were, first, to assess the contribution of a biostatistician
as co-author on the quality of reporting and methodological strength in health research publi-
cations; second, to identify dimensions of reporting quality and study types with methodologi-
cal knowledge gaps; and third, to promote the awareness of the importance of good reporting
among clinical researchers and biostatisticians.

## Materials and methods

### Study design

The study is a retrospective, single-center observational cohort study, conducted at the Univer-
sity of Zurich (UZH) and its University Hospital (USZ).

## Selection of exposed and non-exposed publications

In this study, two groups of publications were compared. The group of "exposed" publications was defined according to their exposure to one or more of a set of 13 academic biostatisticians from the Epidemiology, Biostatistics and Prevention Institute, and the Institute of Mathematics, both localized at University of Zurich, as a co-author. The group will be referred to as biostatisticians in the following. The group of exposed publications was published between 2017 and 2018, and it was retrieved in a PubMed search, with a search string as specified in S1 Appendix on Dec 9, 2019. Methodological publications as well as non-English language publications were excluded.

To define the group of "non-exposed" publications for comparison, all medical research publications found in PubMed between 2017 and 2018, with the affiliation UZH or USZ or any of the affiliated university hospitals for the first and / or the second author were extracted on Dec 16, 2019. Details on the search string can again be found in S1 Appendix. The non-exposed publications have none of the defined set of biostatisticians on the author list. It cannot be excluded that a biostatistician from outside of the group was on the author list. The full list of affiliations considered can be found in S2 Appendix. Based on the full list, a list of affiliations relevant for this study was created, in which for example typographic errors were removed. The large number of non-exposed publications resulting from the affiliation list was used in a random but replicable order—aiming to remove potential chronological ordering or any other systematic ordering while adhering to high standards of reproducibility.

## Categorization into study types

For each of the exposed publications, the study type was determined, and the subset of all RCTs, observational studies, and prediction / prognostic studies was evaluated further. Categorization into study types was performed by the set of biostatisticians. For most publications, the authors themselves determined the study type. For some publications, the biostatistician as co-author had left the department, and thus the study type was categorized independently and in duplicate by two authors (UH, EF). After consensus on study type was reached, record count for each study type for each publication year was obtained. The three study types RCT, observational study, and prediction / prognostic study were the most frequent types. Other types (e.g. systematic reviews) had been abandoned a priori.

The number of non-exposed publications was much larger than the number of exposed publications. For that reason, the categorization of the non-exposed publications into RCTs, observational studies and prediction / prognostic studies was performed in random but replicable order until the numbers of non-exposed publications of these study types matched the corresponding number of exposed publications per year. Categorization was performed independently and in duplicate by the authors UH and MH (for papers published in 2017) and by EF and MH (for 2018). Any discrepancies were resolved by discussion and third-party arbitration (KS). The final set of publications was considered the non-exposed group of publications.

## Selection of items from reporting guideline

For each study type, a set of six items measuring reporting quality were identified by reaching group consensus among the set of biostatisticians. The quality criteria were based on the reporting guidelines CONSORT [7], STROBE, and TRIPOD, and they reflect characteristics of a publication that are especially important for judging the validity of the results and methodological strength.

## Specification of the reporting quality items

The reporting guideline items chosen for the ratings represented the following general dimensions for all three study types: 1. variable specification; 2. how study size was arrived at; 3. missing data; 4. statistical methods; 5. precision of results; and 6. whether the corresponding reporting guideline was mentioned.

The rating of publications regarding these six items was operationalized and piloted, such that they could be used efficiently and robustly to rate each publication consistently. Each dimension had different possible answer categories, also dependent on study type, resulting in a rating varying between 0 (lowest) to 2 (highest) for dimensions 1 to 5, plus an additional point for mentioning the corresponding reporting guideline. Details of the operationalization can be found in S3 Appendix. The range of the total score was from 0 (lowest) to 11 (highest).

## Outcomes

The primary outcome of this study was the sum score of reporting quality and methodological strength in exposed and non-exposed publications, with respect to the six dimensions. The primary outcome was assessed in blinded fashion and in duplicate by two independent raters. The raters were recruited from outside of the departments. Blinding to whether the publication belonged to the exposed or non-exposed group was guaranteed by removing author names, affiliation lists, journal name, corresponding author name, author contributions, date, acknowledgements, references, and DOI from every publication's PDF. Discrepancies in the ratings between the two raters were resolved by a third rating and discussion until consensus was reached.

The secondary outcome of this study was the number of citations in the group of exposed and non-exposed publications at a fixed date (July 20, 2021).

## Outcome rating and rater training

The outcome rating and its operationalization was developed by four authors (UH, KS, MH, EF). After operationalization was finalized the resulting questions for each study type were programmed to be evaluated through an R Shiny app [8], which underwent quality review and a testing period. The questionnaire can be found in S3 Appendix. To find raters, outside of the core study team and outside of the departments, PhD programs in health research across Switzerland, as well as groups of researchers interested in Research on Research were contacted. Each candidate rater could chose a study type, and received written instructions for the rating task. The candidate raters were instructed and trained by rating vignette publications for calibration. These vignette publications of all study types were similar publications as those under study, but they were published in 2019 and were rated with scrutiny by the study authors, including detailed explanation. Only upon successful completion of test ratings, the raters received sets of 11–12 papers of the same study type for rating. The raters were obliged to rate the reporting quality based on the blinded PDF's alone, and not to use additional information from the internet while doing so. Ratings were performed in blinded fashion, meaning that the raters were unaware of the classification of publications as exposed or non-exposed, and of authors on the publications. The ratings were performed in duplicate, and any discrepancies were resolved by a third independent rating. The raters were reimbursed with vouchers for every set of 11–12 publications. Additionally, raters were asked for co-authorship after completion of 33 or more ratings. In total, 15 raters were recruited. The ratings were done between May and July 2021.

## Sample size considerations

The sample size was justified a priori, based on the consideration that with 95 publications in the exposed group, and 95 publications in the non-exposed group at a significance level of 5% and with a power of 80% an effect size of 0.41 (Cohen's $d$) could be detected, using a 2-sided, 2-sample t-test with equal variance assumption. The effect size would be considered a medium effect size. The number of 95 publications corresponded to all publications in the exposed group in the years 2017 and 2018.

## Data management

Data collection in the context of this study had to cover two different aspects. First, categorization of the exposed and non-exposed publications into the three study types was performed with the help of a specifically programmed R Shiny app, in which the title and abstract, as well as the link to the full text was provided, such that the categorization could be performed independently and in duplicate and that any discrepancies could be detected and resolved by discussion. Second, reporting quality rating was performed using another R Shiny app, implementing the operationalized quality dimensions. The electronic records of the two independent ratings, and the consensus rating were saved. The use of R Shiny apps in this research guaranteed highly reliable data entry.

## Risk of bias

The study was designed to compensate the following biases a priori. Risk of detection bias was addressed with blinded and duplicate outcome ratings by researchers not otherwise involved in the study. Risk of selection bias was addressed by considering all publications within two years for the exposed group and by reproducible random sub sampling of PubMed publications from medical publications with UZH / USZ affiliation for the group of non-exposed publications. The results of the study could be confounded by indication, if more complex research projects were brought to the group of biostatisticians' attention whereas less complex projects were addressed by the clinicians without asking for help from an academic biostatistician. This bias was partially addressed by comparing the number of citations of exposed and non-exposed publications, under the hypothesis that equal citation numbers would indicate that less confounding by indication was present.

## Statistical methods and programming

For assessing the level of agreement of reporting quality between the two independent ratings, squared-weighted Cohen's $\kappa$ values were estimated, and reported with 95% confidence intervals based on 1000 bootstrap samples. These analyses of agreement were reported overall and in subgroups of study type. Interpretation of the $\kappa$ values were based on the categorization suggested by Altman [9].

Statistical methods for the primary outcome included visualization of the results with dot-plots (lollipop plots), in which the means of the outcome in the exposed and non-exposed publications are shown, overall (score 0 to 11), and in subgroups of study type (score 0 to 11) and reporting quality dimension (score 0 to 2). Besides that, the estimated between-group differences, overall and in subgroups of study type with 95% confidence intervals (CI) were reported. The two-sided, two-sample t-test under the assumption of equal variances was used to test the hypothesis of no difference in reporting quality between exposed and non-exposed publications. Corresponding Cohen's $d$ was calculated using pooled standard deviations assuming equal variances.

The number of citations was reported overall, and in subgroups of study type, with medians and interquartile ranges, as the distribution was right-skewed. The non-parametric exact Wilcoxon-Mann-Whitney method was used to test the hypothesis of no difference in number of citations between exposed and non-exposed publications, and to estimate a confidence interval. The between-group difference in location was estimated and reported with 95% CI, based on rank statistics.

The software used for statistical analysis was extracted across all publications. It was reported as number and percentage of total. In many publications, more than one software was used, and for that reason the percentages exceed 100%.

All analyses, including subgroup analyses described above were pre-specified in the registered report study protocol [10]. The unit of analysis was the individual publication, or the reporting quality dimension.

Statistical programming was performed with R 4.1.1 [11], in combination with dynamic reporting. Statistical programming included downloading all potential non-exposed publications, random reordering, development of an R Shiny app for categorization of the publications, development of an R Shiny app for the recording of reporting quality ratings, as well as statistical programming of the methods for data analysis and visualization. Results of the study were reported according to STROBE guidelines [12]. All anonymized data was uploaded in an OSF repository.

## Results

In total there were 131 exposed papers published in 2017 and 2018. Of these, 95 publications were of the study types RCT, observational study, or prediction / prognostic study. There were six RCTs, 77 observational studies, and 12 prediction / prognostic studies. The literature search for non-exposed publications with first and / or second author with suitable affiliation and year resulted in a total number of 3420 publications. Four hundred publications of these in random order were categorized into one of the three study types RCT, observational, or prediction / prognostic study, and the retrieved case numbers of the exposed papers could be frequency matched individually for 2017 and 2018. The corresponding flow-chart is shown in Fig 1. All data was made available on OSF [13].

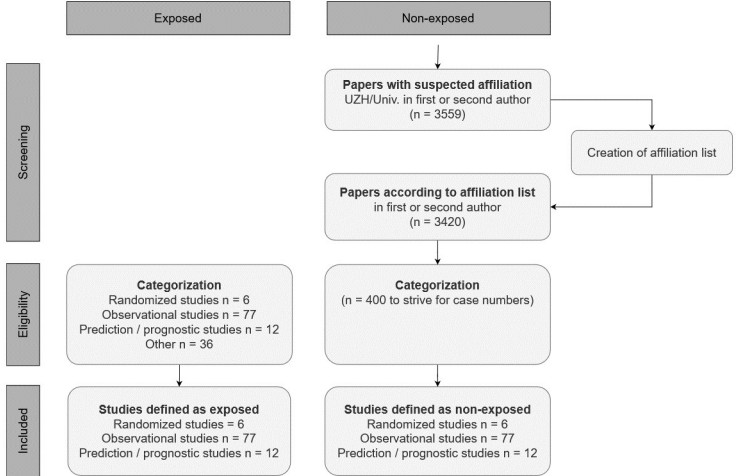

**Fig 1. Flow chart.** Selection process for the exposed publications (left) and the non-exposed publications (right), including screening of affiliation lists of first and second author.

**Table 1. Descriptive statistics.**

|  | Exposed | Non-exposed |
|---|---|---|
| n | 95 | 95 |
| Study type (%) |  |  |
| Randomized Studies | 6 (6.3) | 6 (6.3) |
| Observational Studies | 77 (81.1) | 77 (81.1) |
| Prediction Studies | 12 (12.6) | 12 (12.6) |
| Software used (%) |  |  |
| Excel | 2 (2.1) | 3 (3.2) |
| Graph Pad Prism | 2 (2.1) | 2 (2.1) |
| Matlab | 0 (0.0) | 6 (6.3) |
| Python | 0 (0.0) | 1 (1.1) |
| R | 48 (50.5) | 14 (14.7) |
| SAS | 1 (1.1) | 5 (5.3) |
| SPSS | 38 (40.0) | 40 (42.1) |
| STATA | 15 (15.8) | 12 (12.6) |
| Other software | 2 (2.1) | 5 (5.3) |
| Not mentioned | 11 (11.6) | 21 (22.1) |
| Year (%) |  |  |
| 2017 | 54 (56.8) | 54 (56.8) |
| 2018 | 41 (43.2) | 41 (43.2) |
| Guideline = Mentioned (%) | 10 (10.5) | 2 (2.1) |

Ten of the exposed publications and two of the non-exposed publications mentioned the corresponding reporting guideline. In 48 of the exposed publications, and in 14 of the non-exposed publications, the programming language R was used for the statistical analysis. All descriptive results can be found in Table 1. There were no missing values in the data throughout.

## Agreement

The agreement between the two ratings of each publication was 0.52 (95%CI from 0.46 to 0.57) overall, indicating moderate agreement, according to Altman [9].

For the three different study types, however, the agreement varied between 0.31 (95%CI from 0.05 to 0.52) for RCTs and 0.52 (95%CI from 0.46 to 0.59) and 0.53 (95%CI from 0.35 to 0.68) for observational and prediction studies, respectively. To reach consensus for all ratings with discrepancies a third blinded rater was involved.

## Primary outcome

The estimated between-group difference for the primary outcome was 1.60 (95%CI from 0.92 to 2.28, $p < 0.0001$) in favor of the exposed publications. This result corresponds to a Cohen's *d* of 0.67 (95%CI from 0.38 to 0.97). In the pre-specified subgroups of study type, the estimated between group difference was 3.33 (95%CI from -0.84 to 7.51), 1.39 (95%CI from 0.68 to 2.09) and 2.08 (95%CI from 0.12 to 4.04) for randomized, observational and prediction / prognostic studies, respectively (Fig 2), showing higher reporting quality across all study types. In addition to the estimation of the between group difference, the representation of each subgroup's mean values shows that generally for RCTs the reporting quality was higher than for observational and prediction / prognostic studies.

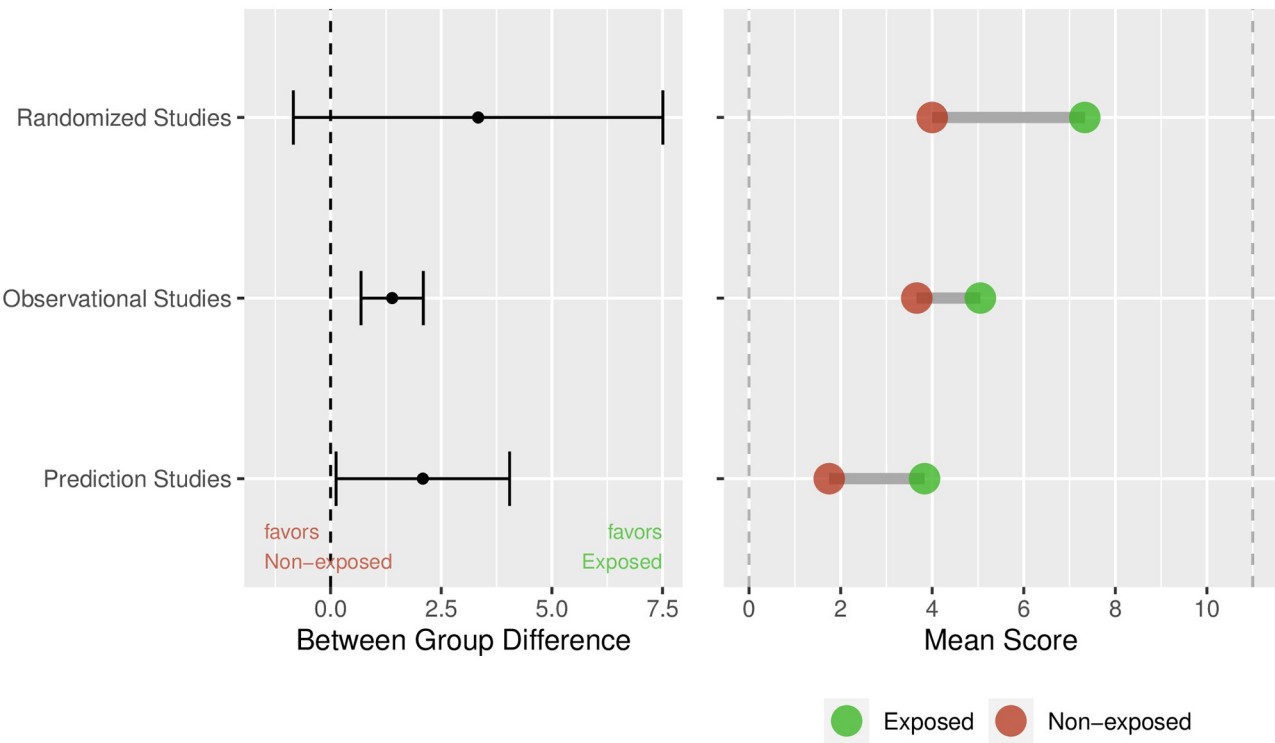

**Fig 2. Estimated between-group difference with 95%CI in the pre-specified subgroups of study type (left); raw means in exposed and non-exposed publications (right).** Unit of analysis is publication.

### Dimension-specific score values

For each of the five reporting dimensions, the between group difference was estimated. The corresponding range of values was between 0 (worst) and 2 (best). Again the results are shown in a graphical representation (Fig 3). The dimension "Variables" had a smaller between group difference, and a higher reporting quality overall, whereas the "Missing data" and "Statistical methods" dimensions were generally reported with less detail. The mean reporting quality in the exposed publications was higher throughout, than that of the non-exposed publications.

### Number of citations

The number of citations, extracted on July 20, 2021, had a non-normal, right-skewed distribution, and for that reason the non-parametric exact Wilcoxon-Mann-Whitney method was used for the estimation of the between group difference and its confidence interval. The estimate was -2 (95%CI from -4 to 0, p = 0.07) indicating weak evidence for higher citation numbers for the non-exposed publications. All descriptive statistics for number of citations can be found in Table 2. It can be seen that the number of citations was relatively balanced for observational studies and prediction / prognostic studies, whereas in the RCTs the number of citations was much larger in the non-exposed group of publications as compared to the exposed publications.

## Discussion

### Summary

Our study demonstrates that the associated effect of academic biostatisticians as co-authors is to increase reporting quality and methodological strength in health research publications,

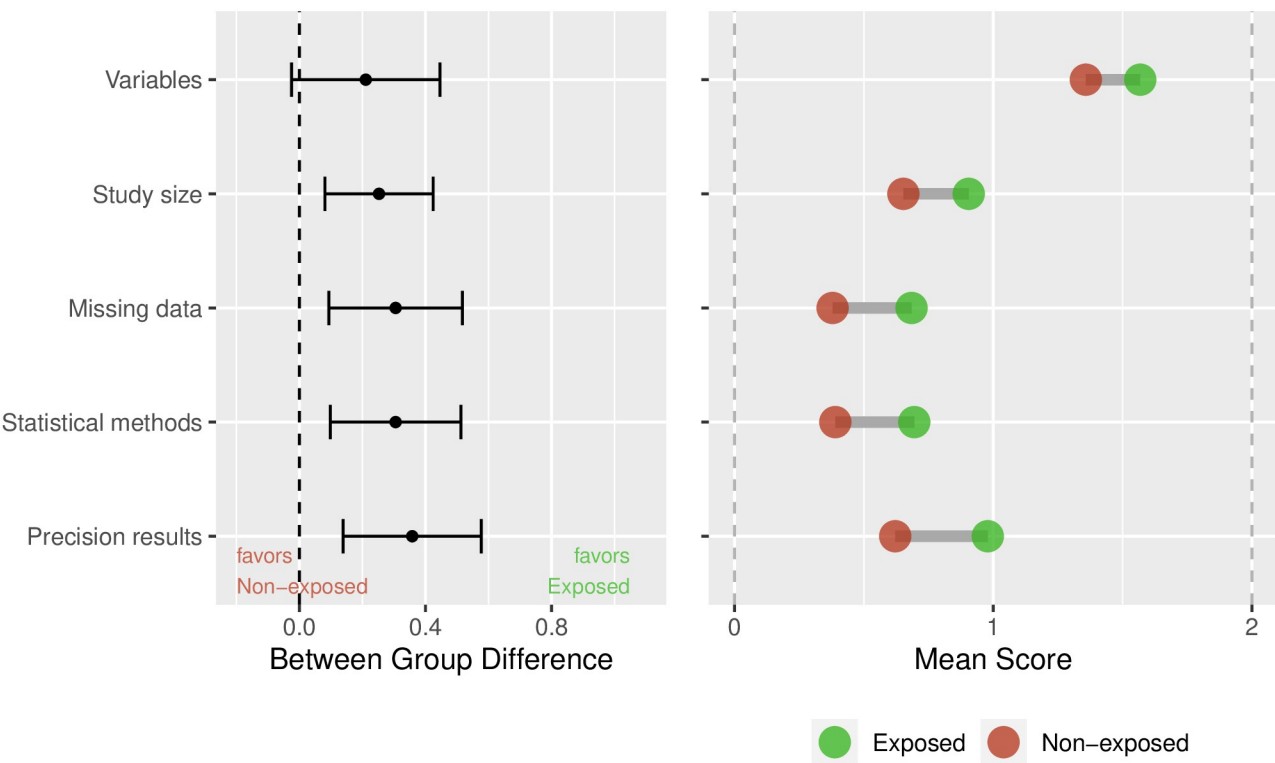

**Fig 3. Estimated between-group difference per dimension with 95%CI (left); raw means in exposed and non-exposed publications (right).** Unit of analysis is dimension.

overall and in subgroups of study types. In addition to that, the subgroup analyses demonstrated that there was evidence for a higher reporting quality in the exposed publications for observational studies and prediction / prognostic studies. The CONSORT statement seems to have been taken up well, because reporting quality was highest generally, for both exposed and non-exposed publications in RCTs.

Citation numbers were comparable for exposed and non-exposed publications in the study types observational and prediction / prognostic studies, but the median number of citations for RCTs was higher in the non-exposed group of publications. The number of citations was evaluated to address the potential bias of confounding by indication. Our findings for observational and prediction / prognostic studies were reassuring, since balanced citation numbers showed that there was no evidence for confounding by indication. The imbalance in citation numbers for RCTs is not necessarily concerning since RCTs may anyhow be considered a special case. They are heavily regulated, CONSORT is generally well enforced by journals, they are expensive studies usually focused on "important" research questions, and they are often

**Table 2. Descriptive statistics for number of citations.** Estimates show the median [IQR].

|  | Exposed | Non-exposed |
|---|---|---|
| Overall | 8.0 [3.0, 14.5] | 9.0 [6.0, 18.5] |
| Randomized Studies | 6.5 [2.0, 10.2] | 35.5 [12.8, 417.5] |
| Observational Studies | 8.0 [3.0, 15.0] | 8.0 [6.0, 17.0] |
| Prediction Studies | 8.5 [6.0, 12.5] | 11.5 [8.0, 17.0] |

multi-center studies and hence likely to be including statisticians from other centers. RCTs are more frequently published in high-ranked medical journals and may therefore have higher citation numbers automatically. Together with the fact that only a low number of RCTs was assessed in this study we believe that there is a low risk of confounding by indication also in the case of RCTs.

Methodological knowledge gaps seemed to be more prominent in the areas of statistical methods, and missing values. Nevertheless the mean reporting quality was higher in the exposed publications, throughout all subgroups. While it seems reasonable to assume that in the exposed papers the biostatisticians knew the methods well, there was still sub optimal reporting of these. The rating of reporting quality was performed in duplicate, and the agreement between first and second ratings were moderate to good, overall. The difficulties in the rating tasks were an indicator of sub optimal reporting in itself. Our study is the first to our knowledge to develop and use a rating score that is usable across study types, and which allows the comparison of reporting quality across study types. Low citation numbers of corresponding reporting guidelines in both, the group of exposed and non-exposed publications may be an indication of lack of awareness among study authors.

## Results in the light of the literature

Our findings are in line with research on research studies evaluating the reporting quality of RCTs, observational studies and prediction / prognostic studies. While the development of reporting guidelines has been ongoing over the last 20 years, and the use of CONSORT is well-established for RCTs, there seem to remain areas in which good reporting less frequently observed. The Cochrane collaboration has initiated the "Prognosis Methods Group" to encourage and facilitate the systematic review and meta analysis of prognostic models in clinical research. Similar to the many systematic evaluations of research questions addressed with RCTs in Cochrane, the field of prediction / prognostic research will benefit and reporting as well as methodological quality will likely increase. Currently, there are 12 prognostic model reviews being undertaken in different fields of clinical research, of which one has been published [14].

Observational studies were the most frequent study type in the sample at hand, and reporting as well as methodological quality was only moderately higher in the exposed publications than in the non-exposed publications. Although the STROBE reporting guidelines have been published in 2004, and taken up by many journals, study authors need continuing reminders as a recent publication in JAMA Surgery by Brooke et al. showed [15].

## Limitations and strengths

Our study has several limitations. The sum score to assess reporting quality and methodological strength was derived and used for the first time in the context of this study. We took multiple means to propose a consistent and valid sum score by using items from the corresponding reporting guidelines CONSORT, STROBE, and TRIPOD directly, and thorough piloting and testing. The sum score addressed the reporting quality in dimensions in which biostatisticians play a relatively prominent role. However, methodological strength could not be rated explicitly. In our view, the assessment of methodological quality is hampered if the reporting quality is low, making reporting assessment and improvement a first important milestone in improving the quality of health research in general. In the selection process, reporting items were chosen that would partially allow the assessment of methodological quality. For example in the dimension "Study size', post-hoc power calculations for observational studies were assigned zero points, and in "Statistical methods" for prediction / prognostic studies zero points were

assigned if model performance measures like discrimination or calibration were not reported. Questions across study types addressed the question of "Missing values", and explicitly asked for methods to address them if present. Risk of bias assessment could be facilitated if reporting quality was higher generally. Another limitation of our study was the low agreement between ratings for the RCTs, which turned out to be only fair. An explanation for this could have been the small number of RCTs being rated by only two different raters. Both raters were relatively consistent in their ratings: one of them being somewhat strict, and the other one relatively lenient. The discrepancies led to the fact that many questions had to be rated by a third rater to come to a consensus.

Our study has several strengths. First of all, we had written a clear study protocol, receiving an external review as a registered report. Upon review of the protocol, the study design and operationalization could be revised and improved. Second, several measures were taken to compensate for different sources of bias, as our study was observational and retrospective. These included double ratings of reporting quality, unbiased assessment of reporting quality through blinded PDFs, and highly reliable data entry through the specifically designed R Shiny app.

### Implications

Our study has several implications for future research. First of all, the study design can repeatedly be applied for future assessments of reporting quality in our group or other academic centers over time. The continuing discussions about the assessment already had an impact on the awareness of the topic among the people involved. In addition, the setup can be generalized to address other documents, e.g., systematic reviews (based on PRISMA [16]), statistical analysis plans [17], or research proposals (SPIRIT [18]).

Academic biostatisticians should take more responsibility in the review of final manuscript versions, and verify the adherence to established reporting guidelines. For reporting of statistical methods and of results with precision, there should be left enough room in the publication. More emphasis should be put on adequate methods to deal with missing values and the reporting thereof.

### Conclusions

Our study is the first to systematically assess the valuable impact of a biostatistician on reporting quality and methodological strength in health research. Higher reporting quality persisted through subgroups of study types and dimensions. The operationalization of the quality assessment allows the direct comparison across study types and dimensions. Methodological knowledge gaps were identified for prediction and prognostic studies, and for the reporting on statistical methods and missing values.

### Supporting information

**S1 Appendix. Search string.** Search string for the identification of potential control publications.
(PDF)

**S2 Appendix. Affiliation list.** Affiliation list for control publications.
(PDF)

**S3 Appendix. Questionnaire.** Questionnaires for the assessment of reporting quality.
(PDF)

## Acknowledgments

We would like to thank Julia Braun, Sarah Haile, and Leonhard Held for their support in extracting relevant items from the corresponding reporting guidelines, and Tina Wünn for support in data preparation and statistical programming of the Shiny apps. We would like to thank the raters involved in our study: Mattia Branca, Victoria Lucia Cammann, Sascha Halvachizadeh, Monika Hebeisen, Brigitta Gahl, Alexandru Garaiman, Andrea Götschi, Stefanie Hayoz, Nadja Hedrich, Arnaud Künzi, Sylvain Losdat, Vera Neumeier, Kelly A. Reeve, Mari Sasaki, Céline Steger, Erin West.

## Author Contributions

**Conceptualization:** Ulrike Held, Klaus Steigmiller, Kelly A. Reeve, Stefanie von Felten, Eva Furrer.

**Data curation:** Ulrike Held, Klaus Steigmiller, Victoria L. Cammann, Alexandru Garaiman, Sascha Halvachizadeh, Sylvain Losdat, Erin Ashley West, Kelly A. Reeve, Eva Furrer.

**Formal analysis:** Ulrike Held, Klaus Steigmiller, Eva Furrer.

**Funding acquisition:** Eva Furrer.

**Investigation:** Klaus Steigmiller, Michael Hediger, Victoria L. Cammann, Alexandru Garaiman, Sascha Halvachizadeh, Sylvain Losdat, Erin Ashley West, Martina Gosteli, Kelly A. Reeve, Stefanie von Felten.

**Methodology:** Ulrike Held, Klaus Steigmiller, Martina Gosteli, Kelly A. Reeve, Stefanie von Felten, Eva Furrer.

**Project administration:** Ulrike Held, Klaus Steigmiller, Michael Hediger, Eva Furrer.

**Resources:** Ulrike Held, Klaus Steigmiller, Victoria L. Cammann, Alexandru Garaiman, Sascha Halvachizadeh, Sylvain Losdat, Erin Ashley West, Martina Gosteli.

**Software:** Ulrike Held, Klaus Steigmiller, Eva Furrer.

**Supervision:** Ulrike Held, Eva Furrer.

**Validation:** Ulrike Held, Klaus Steigmiller, Michael Hediger, Kelly A. Reeve, Stefanie von Felten, Eva Furrer.

**Visualization:** Ulrike Held, Klaus Steigmiller.

**Writing – original draft:** Ulrike Held.

**Writing – review & editing:** Ulrike Held, Klaus Steigmiller, Michael Hediger, Victoria L. Cammann, Alexandru Garaiman, Sascha Halvachizadeh, Sylvain Losdat, Erin Ashley West, Martina Gosteli, Kelly A. Reeve, Stefanie von Felten, Eva Furrer.

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
