## [Decision Letter · Decision Letter 0]

10 Jan 2022

PONE-D-21-37986The incremental value of the contribution of a biostatistician to the reporting quality in health research - a retrospective, single center, observational cohort studyPLOS ONE

Dear Dr. Held,

Thank you for submitting your manuscript to PLOS ONE. After careful consideration, we feel that it has merit but does not fully meet PLOS ONE’s publication criteria as it currently stands. Therefore, we invite you to submit a revised version of the manuscript that addresses the points raised during the review process.

We look forward to receiving your revised manuscript.

Kind regards,

Dylan A Mordaunt, MB ChB, FRACP, FAIDH

Academic Editor

PLOS ONE

Journal Requirements:

"I have read the journal's policy and SL has the following competing interests: SL is employed by CTU Bern, University of Bern, which has a staff policy of not accepting honoraria or consultancy fees. However, CTU Bern is involved in design, conduct, or analysis of clinical studies funded by not-for-profit and for-profit organizations. In particular, pharmaceutical and medical device companies provide direct funding to some of these studies. For an up-to-date list of CTU Bern’s conflicts of interest: http://www.ctu.unibe.ch/research/declaration of interest/index eng.html.

UH, KS, MH, VLC, AG, SH, EAW, MG, KAR, SVF, and EF declare to have no conflict of interest."

Additional Editor Comments:

Thank you for your submission. With regards to the criteria for publication:

1. The study appears to present the results of original research.

2. Results reported don't appear to have been published elsewhere.

3. Experiments, statistics, and other analyses are performed to a high technical standard and are described in sufficient detail. Minor suggestions are made by the reviewers.

4. Conclusions are presented in an appropriate fashion and are supported by the data. Minor suggestions are made.

5. The article is presented in an intelligible fashion and is written in standard English.

6. The research meets all applicable standards for the ethics of experimentation and research integrity.

7. The article adheres to appropriate reporting guidelines and community standards for data availability.

Reviewers' comments:

Reviewer's Responses to Questions

**Comments to the Author**

1. Does the manuscript adhere to the experimental procedures and analyses described in the Registered Report Protocol?

If the manuscript reports any deviations from the planned experimental procedures and analyses, those must be reasonable and adequately justified.

Reviewer #1: Yes

Reviewer #2: Yes

2. If the manuscript reports exploratory analyses or experimental procedures not outlined in the original Registered Report Protocol, are these reasonable, justified and methodologically sound?

A Registered Report may include valid exploratory analyses not previously outlined in the Registered Report Protocol, as long as they are described as such.

Reviewer #1: Yes

Reviewer #2: Yes

3. Are the conclusions supported by the data and do they address the research question presented in the Registered Report Protocol?

The manuscript must describe a technically sound piece of scientific research with data that supports the conclusions. The conclusions must be drawn appropriately based on the research question(s) outlined in the Registered Report Protocol and on the data presented.

Reviewer #1: Partly

Reviewer #2: Yes

4. Have the authors made all data underlying the findings in their manuscript fully available?

Reviewer #1: Yes

Reviewer #2: Yes

5. Is the manuscript presented in an intelligible fashion and written in standard English?

Reviewer #1: Yes

Reviewer #2: Yes

6. Review Comments to the Author

Please use the space provided to explain your answers to the questions above. (Please upload your review as an attachment if it exceeds 20,000 characters)

Reviewer #1: 1. Abstract, Background: “...the contribution of a biostatistician to the reporting and methodological quality..” This idea that the study can determine the biostatistician’s contribution or impact on reporting quality is probably incorrect. This is making a causal assumption where there is only an association. If papers by study teams that include a biostatistician differ on any outcome from papers without a biostatistician among the authors, this can occur for a variety of reasons. One possibility is that the biostatistician made suggestions or wrote sections of the study plan and manuscript, that led to differences in reporting or conduct of the study. There are many other possible explanations, involving differences on the study teams with biostat that led them to include a biostatistician on the team. In other words, background covariates may differ between groups, and these are not measured or controlled for. This causal phrasing is pervasive throughout the manuscript, and I believe it should be removed. For example the first sentence in Discussion (“Our study demonstrates that academic biostatisticians as co-authors have a positive impact on reporting quality and methodological strength in health research publications.. “) should probably be re-written.

2. The difference in number of references in RCTs between groups is not discussed, but seems important. Obviously, study teams without a biostatistician doing RCTs seem to be doing something right. (Note: I notice that the authors are not drawing the conclusion that biostatisticians make RCTs less influential. This would obviously be a wrong interpretation of this finding, since associations cannot be interpreted causally – as discussed in point #1 above).

3. Table 1: I’m curious why the authors have a category for software use, and why R is the only option. I would be interested to know the numbers using SAS as well.

4. Using different pairs of raters for each type of study (RCT, observational, prediction) makes it hard to compare study types on their average compliance with reporting guidelines, since we don’t know if the raters of different study types had the same strictness. So it is not clear if any difference between study types is due to study characteristics or rater characteristics. For example, did RCTs have higher mean scores than observational studies because one of the RCT raters was extremely lenient (rating everything as ‘present’)?

5. Supplemental Table S3 is very helpful for showing how the scores were calculated!

Reviewer #2: See attached files with notes taken during the reads, review and analyses. In general, this work is important to the fields and actions related to meta-analyses, and improving data quality and exploration. The variations seen between different studies of the same topic(s), as a central data resource worker for these research group, offers me the advantage of seeing or predicting the better approaches, and seeing their performance in terms of findings and publication. Applying that experience to this review, I notice I have less understanding of data at the ends of the facilities and researchers engaged with this particular study, and so evaluating people and their performance in the way these authors are doing this, requires more information about the process in general for this research. The purpose of this kind of work should be to encourage others to follow through with the same, and take the same paths, for which reason a little more information on this process is requested for the reader, since the reader can be too self limiting to learn the authors' new method without being able to understand it completely enough. So, in the criticisms I request a few additions, but focus on how to allow future readers to make the best use of understanding the author's thinking and processes for engaging in their analyses. For all of my corporation related work, I do the same myself; yet, it is never required. It is perhaps a little too old fashioned now to, as a statistician, wish to keep your secrets on how you accomplished something to yourself. Your logic should be: for everything I "discover", I have to realize and wonder how many else are out there right now, discovering the same thing. So we, as discoverers, publish our discoveries with the hopes of being recognized as one of those first discoverers, if that is what we are searching for out of our work. The exploratory nature--liking to combine the different methods to produce an more effective, clearer reporting, better research method, is always a better way to explore, than to just reiterate something discovered, without advancing its uses further. And every time a major step is taken, my hope is these researchers are already working on the next generation of this discovery of theirs. In general, working internally, I have found its feels "safest" releasing your novel products when you are two or more levels of advancement, above what you are ready to release. I focus a lot on little things, in relation to overall use, and may be overreacting to the lack of certain items in small sections of the writing, or recommendations when it comes to additions for appendix, or the inclusion of more figures. What tends to be lacking in this field of analytics is spatial thinking and representation, so in many writings, I am frustrates when there are just lists of numbers and flat values, especially if I can't visually analyze them. But unfortunately, the publisher limits their writer's ability to provide their readers with possible multidimensional reviews of what is being discovered. Thus the final product becomes limiting, it seems, and thus people like me asking for more visualization, so that me and others who looks at numbers this way benefit just as much as the table viewer and "memorizer." That being said, this starts up a new line of reasoning, and recommends are more active way of integrating new ideas be taken to produce more helpful and thorough final results in medical research. The authors state, this is a part of the goal of their research--to make better use of the resources that in an ideal setting should already be there. In the US, the biostatistician is mostly hired as a part timer, on a contract basis--industry loves to hire many analysts, who repeat findings and regurgitate the same results again and again. For every several dozen analysts (or more) there are in the health care industry, there is one true tester and discoverer of all possible relationships that may be there. The more we integrate our numbers, the more we see that most others cannot see, and most likely may never see. In health care, there are many things biostatisticians can find, see and display, that analysts will never think of searching for. I just don't get how it is they don't know these things, that they should know. That certainly slows down our numbers related technological advancements, and the improvements of the health care system in general. So that defines the value of these researchers' work. One day, maybe health will stop being a reactionary, retrospective research field, and engage in true spatial prediction modeling. Just like biostatisticians can tell you more, spatial biostatisticians tell you even more.

7. PLOS authors have the option to publish the peer review history of their article (what does this mean?). If published, this will include your full peer review and any attached files.

Reviewer #1: No

Reviewer #2: **Yes: **Brian L Altonen

---

## [Author Response · Author response to Decision Letter 0]

11 Feb 2022

Point to point reply for manuscript PONE-D-21-37986

The incremental value of the contribution of a biostatistician to the reporting quality in health research - a retrospective, single center, observational cohort study

February 11, 2022

Reply to Reviewer # 1

1. Abstract, Background: “...the contribution of a biostatistician to the reporting and methodological quality..” This idea that the study can determine the biostatistician’s contribution or impact on reporting quality is probably incorrect. This is making a causal assumption where there is only an association. If papers by study teams that include a biostatistician differ on any outcome from papers without a biostatistician among the authors, this can occur for a variety of reasons. One possibility is that the biostatistician made suggestions or wrote sections of the study plan and manuscript, that led to differences in reporting or conduct of the study. There are many other possible explanations, involving differences on the study teams with biostat that led them to include a biostatistician on the team. In other words, background covariates may differ between groups, and these are not measured or controlled for. This causal phrasing is pervasive throughout the manuscript, and I believe it should be removed. For example the first sentence in Discussion (“Our study demonstrates that academic biostatisticians as co-authors have a positive impact on reporting quality and methodological strength in health research publications.. “) should probably be re-written.

Reply: Thank you for the suggestion. We have reworded the manuscript throughout, to avoid a causal interpretation of the findings, and replaced it by terms pointing into the direction of an association.

2. The difference in number of references in RCTs between groups is not discussed, but seems important. Obviously, study teams without a biostatistician doing RCTs seem to be doing something right. (Note: I notice that the authors are not drawing the conclusion that biostatisticians make RCTs less influential. This would obviously be a wrong interpretation of this finding, since associations cannot be interpreted causally – as discussed in point #1 above).

Reply: We agree with the Reviewer that a thorough discussion of the number of citations of RCTs with and without biostatistician was not included. We would like to mention that the citation numbers in our paper were used specifically to address the potential risk of bias due to confounding by indication. We extended the discussion section as requested.

3. Table 1: I’m curious why the authors have a category for software use, and why R is the only option. I would be interested to know the numbers using SAS as well.

Reply: Thank you for bringing this topic up. Our results showed that in many publications, more than one software package was being used, e.g. R and Stata, or SPSS and Stata. Therefore, no simple categories of different software packages that would add up to 100% could be provided. We revised the analysis according to your suggestion and made new categories, showing which software was used (R, Stata, SAS, SPSS, other), potentially in combination with other software. The manuscript was revised to explain the new definition of categories.

4. Using different pairs of raters for each type of study (RCT, observational, prediction) makes it hard to compare study types on their average compliance with reporting guidelines, since we don’t know if the raters of different study types had the same strictness. So it is not clear if any difference between study types is due to study characteristics or rater characteristics. For example, did RCTs have higher mean scores than observational studies because one of the RCT raters was extremely lenient (rating everything as ‘present’)?

Reply: Thank you for this comment. We do not believe that this issue is a concern because we had a large number of different raters across study types, and except for the very small number of RCTs there were no “pairs” of raters across observational studies and prediction / prognostic studies. Please let us clarify that the training to prepare raters to individual study types was very time consuming for the raters and it was not reimbursed by vouchers, so it was a strategic decision to train raters primarily for a specific study type.

5. Supplemental Table S3 is very helpful for showing how the scores were calculated!

Reply: Thank you for this comment.

 

Reply to Reviewer # 2

1. In general, this work is important to the fields and actions related to meta-analyses, and improving data quality and exploration. The variations seen between different studies of the same topic(s), as a central data resource worker for these research group, offers me the advantage of seeing or predicting the better approaches, and seeing their performance in terms of findings and publication. Applying that experience to this review, I notice I have less understanding of data at the ends of the facilities and researchers engaged with this particular study, and so evaluating people and their performance in the way these authors are doing this, requires more information about the process in general for this research. The purpose of this kind of work should be to encourage others to follow through with the same, and take the same paths, for which reason a little more information on this process is requested for the reader, since the reader can be too self limiting to learn the authors' new method without being able to understand it completely enough. So, in the criticisms I request a few additions, but focus on how to allow future readers to make the best use of understanding the author's thinking and processes for engaging in their analyses. For all of my corporation related work, I do the same myself; yet, it is never required. It is perhaps a little too old fashioned now to, as a statistician, wish to keep your secrets on how you accomplished something to yourself. Your logic should be: for everything I "discover", I have to realize and wonder how many else are out there right now, discovering the same thing. So we, as discoverers, publish our discoveries with the hopes of being recognized as one of those first discoverers, if that is what we are searching for out of our work. The exploratory nature--liking to combine the different methods to produce an more effective, clearer reporting, better research method, is always a better way to explore, than to just reiterate something discovered, without advancing its uses further. And every time a major step is taken, my hope is these researchers are already working on the next generation of this discovery of theirs. In general, working internally, I have found its feels "safest" releasing your novel products when you are two or more levels of advancement, above what you are ready to release. I focus a lot on little things, in relation to overall use, and may be overreacting to the lack of certain items in small sections of the writing, or recommendations when it comes to additions for appendix, or the inclusion of more figures. What tends to be lacking in this field of analytics is spatial thinking and representation, so in many writings, I am frustrates when there are just lists of numbers and flat values, especially if I can't visually analyze them. But unfortunately, the publisher limits their writer's ability to provide their readers with possible multidimensional reviews of what is being discovered. Thus the final product becomes limiting, it seems, and thus people like me asking for more visualization, so that me and others who looks at numbers this way benefit just as much as the table viewer and "memorizer." That being said, this starts up a new line of reasoning, and recommends are more active way of integrating new ideas be taken to produce more helpful and thorough final results in medical research. The authors state, this is a part of the goal of their research--to make better use of the resources that in an ideal setting should already be there. In the US, the biostatistician is mostly hired as a part timer, on a contract basis--industry loves to hire many analysts, who repeat findings and regurgitate the same results again and again. For every several dozen analysts (or more) there are in the health care industry, there is one true tester and discoverer of all possible relationships that may be there. The more we integrate our numbers, the more we see that most others cannot see, and most likely may never see. In health care, there are many things biostatisticians can find, see and display, that analysts will never think of searching for. I just don't get how it is they don't know these things, that they should know. That certainly slows down our numbers related technological advancements, and the improvements of the health care system in general. So that defines the value of these researchers' work. One day, maybe health will stop being a reactionary, retrospective research field, and engage in true spatial prediction modeling. Just like biostatisticians can tell you more, spatial biostatisticians tell you even more.

Reply: We thank the reviewer for these general comments regarding the role of a biostatistician, in general and specifically. We would like to point to the fact that the level of evidence of “discovery” in our study is high due to the thorough and rigorous writing up of the study protocol and review thereof as registered report on PLOS ONE. While the study was conducted and analysed, we fully adhered to the study protocol.

Publication recommended, with: 

2. expansion of or expounding/clarification of bias sections notes [see lines 145-156]

Reply: Thank you for your comment on risk of bias. The section was revised to comment on biases that could have resulted from our way of selection publications in this study.

3. R Studio resource link: i.e. https://shiny.rstudio.com/ or better—[add to test or appendix] 

Reply: Thank you for bringing this point to our attention. The reference for the shiny package was added to the references of our paper.

4. provide a “visualization” example, flowchart, something for exemplification [see lines 163-165]

Reply: We would like to point to the fact that our study includes a flow chart, which was uploaded as figure 1. Additionally, the lollipop plots are provided as figures 2 and 3.

5. “Table 1. Descriptive Statistics”, a footnote detailing the “other” software products might be worth considering (although not required). [after line 203]

Reply: We would like to thank the reviewer for this comment, it was an issue raised also by Reviewer #1. We added categories of other software packages to the table 1, including Stata, SPSS, SAS, etc.

 

Optionals (for readers):

--Discussion Section. further description, definition, examples of “sub optimal reporting” [254-256]; are there qualifiers defined for this? —[add to appendix]

Reply: The qualifiers for reporting quality are the corresponding items of the reporting guidelines of the selected study types. We used the original reporting guideline items to operationalize the questionnaire of this study. This is described in the methods section, and also in S3- Appendix, in which the detailed Questionnaire is described.

--summary of all qualifiers/quantifiers for 0,1,2 scoring, as used by raters—[add to appendix].

Reply: See above, the information is provided in S3 Appendix.

--rater/review sheet, etc. [preferred, although optional(?)] actual sample of result, or idealized example of result, in format used to keep the notes and observations, either onscreen, in a (excel like) table, on paper, in a notebook, etc.. —[add to appendix]

Reply: The review was performed with an R shiny app. Therefore no paper sheets were used for the rating task. The data was made available on OSF.

--data descriptive metadata file—[add to appendix]

Reply: The data was made available on OSF. We give the link in the manuscript.

---

## [Editor Report · Decision Letter 1]

18 Feb 2022

The incremental value of the contribution of a biostatistician to the reporting quality in health research - a retrospective, single center, observational cohort study

PONE-D-21-37986R1

Dear Dr. Held,

We’re pleased to inform you that your manuscript has been judged scientifically suitable for publication and will be formally accepted for publication once it meets all outstanding technical requirements.

Kind regards,

Dylan A Mordaunt, MB ChB, MPH, MHLM, FRACP, FAIDH

Academic Editor

PLOS ONE

Additional Editor Comments (optional):

Thank you for your resubmission. This now meets the criteria for publication.
---

## [Editor Report · Acceptance letter]

24 Feb 2022

PONE-D-21-37986R1 

The incremental value of the contribution of a biostatistician to the reporting quality in health research - a retrospective, single center, observational cohort study 

Dear Dr. Held:

I'm pleased to inform you that your manuscript has been deemed suitable for publication in PLOS ONE. Congratulations! Your manuscript is now with our production department. 

Kind regards, 

on behalf of

Dr. Dylan A Mordaunt 

Academic Editor

PLOS ONE